# Natural Language Induced Adversarial Images

## ABSTRACT

Research of adversarial attacks is important for AI security because it shows the vulnerability of deep learning models and helps to build more robust models. Adversarial attacks on images are most widely studied, which includes noise-based attacks, image editing-based attacks, and latent space-based attacks. However, the adversarial examples crafted by these methods often lack sufficient semantic information, making it challenging for humans to understand the failure modes of deep learning models under natural conditions. To address this limitation, we propose a natural language induced adversarial image attack method. The core idea is to leverage a text-to-image model to generate adversarial images given input prompts, which are maliciously constructed to lead to misclassification for a target model. To adopt commercial text-to-image models for synthesizing more natural adversarial images, we propose an adaptive genetic algorithm (GA) for optimizing discrete adversarial prompts without requiring gradients and an adaptive word space reduction method for improving the query efficiency. We further used CLIP to maintain the semantic consistency of the generated images. In our experiments, we found that some high-frequency semantic information such as "foggy", "humid", "stretching", etc. can easily cause classifier errors. These adversarial semantic information exist not only in generated images, but also in photos captured in the real world. We also found that some adversarial semantic information can be transferred to unknown classification tasks. Furthermore, our attack method can transfer to different text-to-image models (e.g., Midjourney, DALL·E 3, etc.) and image classifiers.

## CCS CONCEPTS

• **Security and privacy** → *Social aspects of security and privacy*;
• **Computing methodologies** → *Computer vision*; *Bio-inspired approaches*.

## KEYWORDS

Adversarial Example, Adversarial Attack, Text-to-Image model, Social Aspects of Generative AI, Vision and Language

## 1 INTRODUCTION

As widely acknowledged, some carefully designed inputs called adversarial examples can mislead the deep learning models. The perturbation process is called adversarial attack [3, 17, 41]. Adversarial attacks can identify the vulnerability of deep learning models, and facilitate the development of more robust models. Currently,

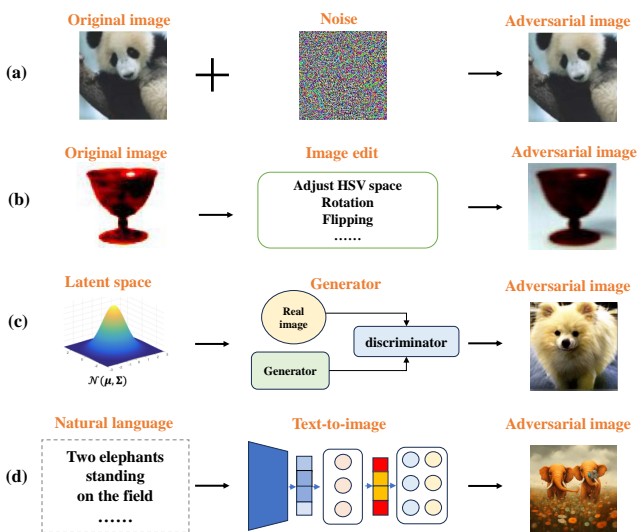

**Figure 1: Different adversarial image attacks. (a) Noise-based attack. (b) Image editing-based attack. (c) Latent space-based attack. (d) Natural language induced adversarial image attack (Ours).**

most adversarial attacks focus on adversarial images, which can be roughly categorized into three types (Figure 1).

The first type is noise-based attack [3, 17, 35, 41, 59], which generates adversarial examples by adding adversarial noise to the image. The second type is image editing-based attack [62, 63, 66], which modifies certain properties (e.g. HSV, brightness, etc. ) of the image. The third type is latent space-based attack [30, 64]. This attack guides the generators such as GAN to generate adversarial images by modifying the latent space variables of the generators.

If we want to understand under what natural conditions images are easily misled in classification, the above methods are ineffective because they are difficult to incorporate semantic information during attacks. To describe natural situations, the most convenient method for users is language. For example, users can use language to depict numerous natural scenes (such as various weather conditions or different gestures of objects), utilize text-to-image models to generate a large number of images, and test an image classifier on which natural scenarios it is easy to be misled.

To achieve this goal, we propose a natural language induced adversarial image attack method. Language is one of the easiest ways to be understood by humans. The current progress in text-to-image models [42, 48] makes it possible for us to use natural language to generate adversarial images according to our needs. The core idea is to leverage a text-to-image model to generate adversarial images given input prompts, which are maliciously constructed to lead to misclassification for a target model. We construct the adversarial prompts by optimizing the words in prompts. Our language-based

method has rich semantic information and helps humans to analyze the adversarial images from a natural language view.

Optimizing the words in prompts for text-to-image models faces challenges. First, each word in a sentence is a discrete variable, which is difficult to be optimized using gradient-based methods. Second, many commercial text-to-image models such as Midjourney are black-box models whose gradients and parameters are not accessible. Third, some commercial models such as DALL·E 3 limit the number of queries, which bring difficulty for the adversarial optimization. Besides, we should make the generated images contain enough semantic information consistent with the prompts during the optimization.

To adopt commercial text-to-image models for synthesizing more natural adversarial images, we propose an adaptive genetic algorithm (GA) for optimizing discrete adversarial prompts without requiring gradients and an adaptive word space reduction method for improving the query efficiency. We further used CLIP to maintain the semantic consistency of the the generated images.

We evaluated our method on different classification attack tasks. In our experiments, we found that some high-frequency semantic information such as "foggy", "humid", "stretching", etc. can easily cause classifier errors. These adversarial semantic information exist not only in generated images, but also in photos captured in the real world. We also found that some adversarial semantic information can be transferred to unseen classification tasks. Furthermore, our attack method can transfer to different text-to-image models (e.g., Midjourney, DALL·E 3, etc.) and image classifiers. Our method helps people to better understand the weakness of classifiers from a natural language perspective. Through experiments, we also reveal the potential safety and fairness issues of current text-to-image models. It inspires us to build more robust and fair AI models.

## 2 RELATED WORKS

### 2.1 Noise-Based Attacks

These attacks generate adversarial images by adding adversarial noises on the original images. Classical methods include L-BPGS [52], FGSM [17], PGD [41], C&W [3], etc. Some recent works further improved the strength and feasibility of noise-based attacks. For example, SparseFool [43], ADMM [61] and LP-BFGS [67] enhanced the group sparsity of perturbations. PONS [19], HO-FMN [16] and FAB-Attack [11] maintained attack performance with less computational efforts during noise searching. Xie [60], Rahmati [46], Ilyas [25] and Ergezer [14] generalized noise-based attack to new scenarios, such as anchor-free detectors, multi-angle detectors, black-box models, etc.

### 2.2 Image Editing-Based Attacks

These attacks operate image transformations to generate adversarial images. The early works [13, 20, 29, 62] mainly involved image rotation, flipping, and adjustment of the HSV space. Some recent works introduced more complex image processing methods. For example, Liu [33], Zeng [65] used additional differentiable renderers to do image transformations. Wang [57] leveraged perception similarity supervision [68] to enlarge adversarial perturbations.

## 2.3 Latent Space-Based Attacks

These attacks change the latent space of generative models to generate adversarial images. Zhao [69], Lin [32], Hu [22], Lapid [30] and Lau [31] used Generative Adversarial Network (GAN) to generate adversarial images by finetuning its generator. Xue [64], Wang [55], Chen [4], Liu [34] and Chen [6] used diffusion models to generate adversarial images by optimizing the parameters of the U-Net structure, or by adding learned noises in the latent space.

## 2.4 Text-to-Image Models

Text-to-image models are a group of multimodal generative models that can create images from text prompts. These models firstly encode the text prompt into a latent space, then circularly and conditionally denoising a Gaussian Distribution back to an image. The denoising process are trained from a predefined forward process. Influential Text-to-image models include Midjourney[42], Stable Diffusion[48], DALL·E 2 [47], Imagen [49], etc.

## 3 METHODS

### 3.1 Problem Formulation and Overview

Our idea is to optimize the words within a sentence to obtain prompts for text-to-image models, and then input the prompts to text-to-image models to obtain adversarial images. Let $W$ denote the word space, including subjects, verbs, adjectives, etc. These words can be combined into a prompt $p$ according to grammatical order. Let Combination denote this function. Let $G$ denote the text-to-image model. For any prompt $p$, $G(p)$ is the generated image with the ground truth category $y$. Let $f$ denote the image classifier. Our goal is to conduct an untargeted attack, and we hope that by optimizing $p$, the classifier $f$ will misclassify the image $G(p)$ into a category other than $y$. We define the attack success rate of $p$ as ASR$(p)$. At the same time, we hope that the generated image $G(p)$ contain enough target semantic information of ground truth category $y$. For this purpose, we define the target semantic information strength as SEM$(p)$. We formulate the problem as:

$$\begin{aligned} \underset{p}{\text{maximize}} \quad & \text{ASR}(p) + \lambda \cdot \text{SEM}(p) \\ \text{subject to} \quad & p = \text{Combination}(W), \end{aligned} \quad (1)$$

where $\lambda$ is determined empirically.

To optimize the prompt $p$, we propose an adaptive genetic algorithm for optimizing discrete adversarial prompts without requiring gradients and an adaptive word space reduction method for improving the query efficiency. We further used CLIP to maintain the semantic consistency of the the generated images. The overal pipeline of our method is shown in Figure 2.

### 3.2 Building the Word Space and Prompts

The adversarial prompt structure is customizable. For example, in our animal classification attack experiments, the prompt structure is defined as

"*<number><color>[target animal] <appearance>is <gesture>on the <background>on a <weather>day, the [target animal] faces forward, the [target animal] occupies the main part in this scene, viewed <viewangle>.*"

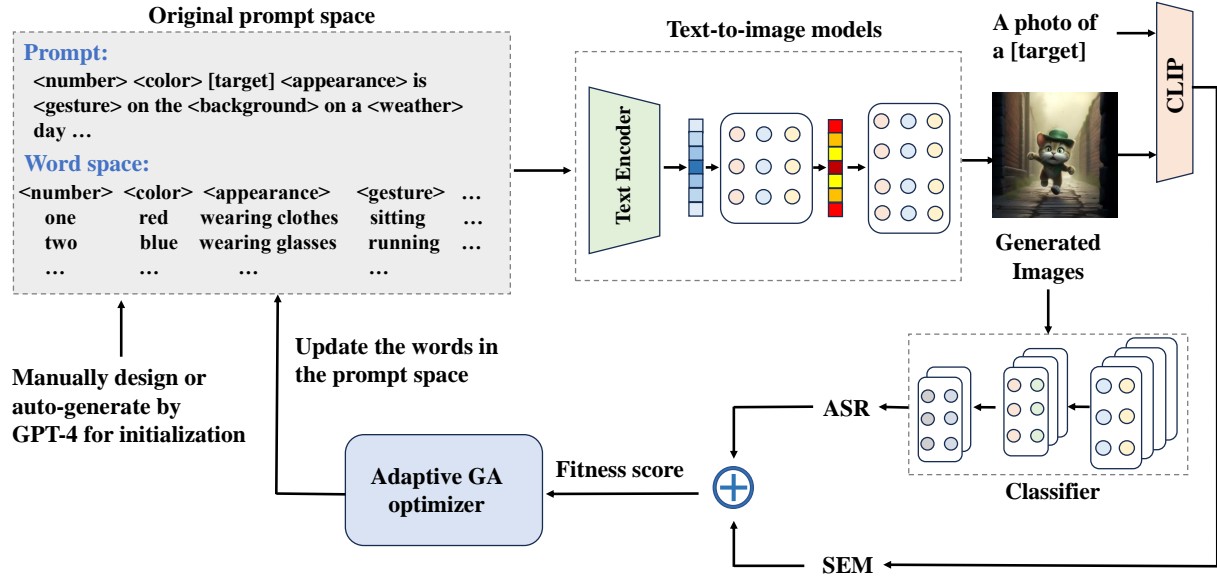

**Figure 2: The overall pipeline of the proposed method.**

The optimization word space is also customizable. *"<word>"* represents a word that can be optimized. For example, in our experiments, the word space of *"<weather>"* is *{ "sunny", "rainy", "cloudy", "snowy", "windy", "foggy", "stormy", "humid" }*. *"[target animal]"* is the ground truth target category $y$ (e.g. "cat") of the generated images, which is user-defined in prompt $p$ and fixed during the prompt optimization.

We can also use GPT-4 to automatically construct the word space, which can be transferred to other classification tasks. Here are the steps: First, we can select a target category, such as race, vehicle, etc. Next, the above hand-constructed word space is input into GPT-4 as an example, and GPT-4 is instructed to generate a similar word space for new tasks. Details are introduced in *Supplementary Material (SM)*.

The settings of other prompts and word spaces are introduced in *SM*. The word space $W$ and the set of prompts $P$ are formulated as follows:

$$W = \{w_1, w_2, ..., w_M\},$$
$$P = \{p_1, p_2, ..., p_N\}. \tag{2}$$

where

$$p_i = \text{Combination}(W), i = 1, 2, ..., N. \tag{3}$$

### 3.3 Fitness Evaluation

We optimize the adversarial prompts based on genetic algorithm which simulates the genetic evolution process of a population. We assume that there are $N$ prompts, constituting a population $P$, and each prompt $p$ is an individual in this population. One critical task is to evaluate the fitness of these individuals, simulating the natural selection process to retain the most optimal individuals. The fitness function $\mathbb{F}$ is designed according to the Equation 1, which is

$$\mathbb{F}(p) = \text{ASR}(p) + \lambda \cdot \text{SEM}(p). \tag{4}$$

*3.3.1 ASR.* To evaluates the attack performance of our method, we define the attack success rate (ASR) as the ratio of the number of successfully attacked images generated by the text-to-image model $G$ using prompt $p$, denoted as $N_{f(G(p))\neq y}$, to the total number of generated images, denote as $N_{G(p)}$. The calculation formula is

$$\text{ASR}(p) = N_{f(G(p))\neq y}/N_{G(p)}. \tag{5}$$

*3.3.2 SEM.* Our goal is to generate adversarial images that contain enough target semantic information consistent with the prompts. One challenge is how to to maintain the semantic consistency of the the generated images. To address this issue, we employ the CLIP [44] model's text Encoder $E_T$ and image Encoder $E_I$ to calculate the cosine distance between the generated image $G(p)$ and the target semantic information $g_t$ of ground truth category $y$ (e.g., "a photo of a cat"). This measure reflects their relevance, considering CLIP's robust multimodal capabilities, enabling accurate assessment of the semantic correlation between the image content and the target semantic text. Besides, CLIP is trained on a large-scale (i.e. 400 million) dataset, exhibiting strong generalization across diverse image styles and backgrounds. To enhance the target semantic information in adversarial images, we incorporate it as part of the fitness function during the genetic optimization process, specifically as

$$\text{SEM}(p) = \frac{E_I(G(p)) \cdot E_T(g_t)}{\|E_I(G(p))\|_2 \cdot \|E_T(g_t)\|_2}. \tag{6}$$

### 3.4 Adaptive Word Space Reduction

The number of queries is closely related to optimization time and cost of using commercial text-to-image models. Besides, some models such as DALL·E 3 limit the number of queries. To reduce the number of queries, we propose an adaptive word space reduction method. The core idea is to select the individual with the lowest fitness, denoted as $p_{lowest}$, in each generation. Two words, $w_{attr1}$ and $w_{attr2}$, are randomly chosen from $p_{lowest}$, and these two words are

removed from the word space. This is similar to eliminate weaker genes from the gene pool based on fitness in the current generation $t$, retaining relatively high-quality genes for the next $t + 1$ generation's reproduction, that is

$$W^{(t+1)} = \text{AdaptiveReduce}\left(W^{(t)}, w_{\text{attr1}}, w_{\text{attr2}}\right). \quad (7)$$

## 3.5 Optimization of Adversarial Prompts

We optimize the adversarial prompts based on GA algorithm, the optimization process includes prompts initialization, crossover, mutation, selection, iteration and termination.

*3.5.1 Prompts Initialization.* We initialize $N$ prompts $P_{init}$ by randomly selecting words from word space. These prompts can be regarded as parent prompts, which are candidates for evolution.

*3.5.2 Crossover.* The crossover operation is to select two parent prompts $P_{\text{parent1}}$, $P_{\text{parent2}}$ each time to generate child prompts $P_{\text{child}}$ by exchanging words. Different from the standard GA algorithm that randomly selects parents with a fixed probability, we set the probability $pc$ of selecting each prompt as a parent is proportional to its fitness score as shown in Equation 8, assuming that parents with higher fitness are more likely to produce offspring with higher fitness. Each word is like a gene, and the offspring randomly selects the genes of either parent.

$$pc = \frac{\mathbb{F}\left(p_i^{(t)}\right)}{\sum_{j=1}^{N} \mathbb{F}\left(p_j^{(t)}\right)}, 1 \le i, j \le N. \quad (8)$$

$$P_{\text{child}} = \text{Crossover}\left(P_{\text{parent1}}, P_{\text{parent2}}, pc\right). \quad (9)$$

*3.5.3 Mutation.* During the evolution of a population, mutations may occur in the genes of individuals, which contributes to the diversity of the population. Similar to this biological process, we set a small probability $pm$ for each word in a prompt to be randomly changed to another word of the same type. This helps us avoid local optimal solutions. The new population with mutated individuals are

$$P_{\text{mutated}} = \text{Mutation}\left(P_{\text{child}}, pm\right). \quad (10)$$

*3.5.4 Selection.* We use a roulette strategy to select prompts for the next generation. This means that the probability $ps$ of each offspring surviving is proportional to their fitness, and is calculated using the Equation 11. In this way, we select individuals with highest fitness, reflecting the natural principle of "survival of the fittest" in the evolutionary process. So

$$ps = \frac{\mathbb{F}\left(p_i^{(t+1)}\right)}{\sum_{j=1}^{N} \mathbb{F}\left(p_j^{(t+1)}\right)}, 1 \le i, j \le N. \quad (11)$$

$$P_{\text{selected}} = \text{Selection}\left(P_{\text{mutated}}, ps\right). \quad (12)$$

*3.5.5 Iteration and Termination Condition.* The crossover, mutation, and selection are performed iteratively. There are two iteration termination conditions: one is when the number of iterations reaches a threshold $\alpha$, and the other is when the success rate reaches a threshold $\beta$. After the termination, the final batch of retained offspring prompts serves as the set of adversarial prompts. These prompts are then fed into the text-to-image model to generate adversarial images.

## 4 EXPERIMENTS

### 4.1 Text-to-Image Models

We mainly used the Midjourney [42], which is a powerful commercial text-to-image model to generate the natural language induced adversarial images. We also tested our method on the other famous text-to-image models including DALL·E 2 [47], DALL·E 3 [2], Stable Diffusion [48], Mysterious XL v4 [10], Dreamshaper XL alpha 2 [8], and Real Cartoon XL v4 [9].

### 4.2 Dataset

*4.2.1 ImageNet.* ImageNet is one of the largest publicly available datasets for image classification tasks, consisting of over 14 million images annotated with around 22,000 categories. The target classifiers in our experiments were pre-trained on ImageNet. For classification attacks, we selected 10 animal categories from ImageNet as the target categories, which was the same as those of Animal-10 [1] dataset.

*4.2.2 Animals-10.* Due to the category imbalance in ImageNet (e.g. "dog" contains 118 sub-categories with 148,418 images, while "horse" only contains 1 sub-categorie with 1300 images), which may cause unbalanced classification performance and attack effects for different categories, as detailed in Section 4.5. Therefore, we chose a category-balanced dataset Animals-10 [1] released in the Kaggle platform. It contains around 28,000 animal images which belongs to 10 categories: cat, dog, spider, horse, chicken, butterfly, cow, sheep, elephant, squirrel. This dataset is used to finetune the animal image classifiers, which were pre-trained on ImageNet.

*4.2.3 FairFace.* We used the FairFace dataset [26] which contains 108,501 images balanced on race. It includes 7 groups: Black, White, East Asian, Middle Eastern, Southeast Asian, Indian and Latino. This dataset is used to finetune the race image classifier, which were pre-trained on ImageNet.

### 4.3 Target Classifiers

For animal image classifiers, we used the models including ResNet [18], ViT [12], VGG [50], Inception v3 [51], DenseNet [23], MobileNet [21], EfficientNet [53], SqueezeNet [24], RegNet [45], AlexNet [27] implemented in the torchvision library. We also used two adversarial trained models: Swin-L [38] and ConvNeXt-L [39]. For race image classifier, we used the ViT model. The accuracy of the finetuned classifiers are all above 98% on the corresponding dataset.

### 4.4 Evaluation Metrics

We used the attack success rate (ASR) as the evaluation metric for our attack method, which is widely used by previous works [5, 36, 37, 58]. The ASR is defined as the ratio of misclassified images to the total number of generated images. Its calculation method has been introduced in Section 3.3.1.

One blue dog wearing clothes is taking a nap on the moon ……

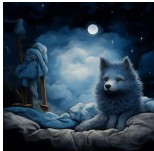

Dog ⟶ Cat

One purple cat wearing a flower on the head is……

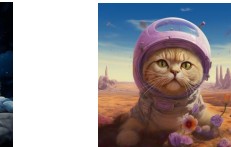

Cat ⟶ Butterfly

One orange sheep wearing a flower on the head ……

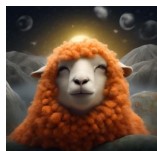

Sheep ⟶ Dog

One yellow horse wearing a pair of glasses is crawling ……

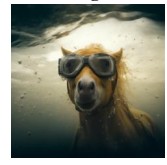

Horse ⟶ Dog

One yellow chicken wearing a pair of glasses is ……

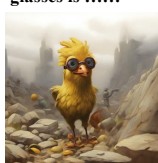

Chicken ⟶ Dog

One purple elephant wearing a hat is taking a nap ……

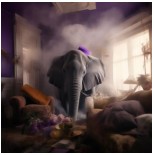

Elephant ⟶ Cat

One yellow cow wearing a pair of glasses is sitting ……

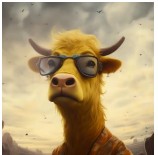

Cow ⟶ Sheep

One blue butterfly wearing clothes is sitting ……

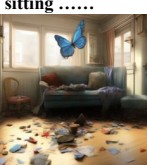

Butterfly ⟶ Cat

One brown squirrel wearing a pair of glasses is bark……

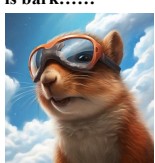

Squirrel ⟶ Cat

One white spider wearing a pair of glasses is sit ……

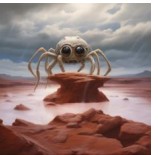

Spider ⟶ Squirrel

**Figure 3: Examples for animals classifier attacks. The black texts are the prompts, the blue texts are the groundtruth categories, and the red texts are the misclassified categories.**

**Table 1: ASRs (%) of different methods against animal classifiers trained on ImageNet. M: methods. T: target animal**

| M \ T | Sheep | Dog | Cat | Horse | Cow | Chicken | Elephant | Butterfly | Spider | Squirrel | Average |
|---|---|---|---|---|---|---|---|---|---|---|---|
| Clean | 0.0 | 29.2 | 5.8 | 0.0 | 0.8 | 0.0 | 0.0 | 5.0 | 0.0 | 0.0 | 4.1 |
| Random | 56.5 | 76.5 | 44.8 | 54.5 | 22.3 | 3.0 | 15.0 | 31.0 | 18.8 | 42.3 | 36.5 |
| Comb | 60.0 | 78.5 | 37.2 | 47.0 | 40.0 | 2.5 | 17.9 | 31.3 | 22.5 | 39.0 | 37.6 |
| Ours | **83.1** | **89.4** | **78.1** | **95.6** | **78.8** | **77.2** | **80.3** | **92.8** | **86.3** | **85.0** | **84.7** |

**Table 2: ASRs (%) of different methods against animal classifiers finetuned on Animals-10. M: methods. T: target animal**

| M \ T | Sheep | Dog | Cat | Horse | Cow | Chicken | Elephant | Butterfly | Spider | Squirrel | Average |
|---|---|---|---|---|---|---|---|---|---|---|---|
| Clean | 0.0 | 0.0 | 0.0 | 4.2 | 0.8 | 3.3 | 0.0 | 0.0 | 0.8 | 0.8 | 1.0 |
| Random | 44.5 | 12.7 | 15.3 | 22.1 | 41.6 | 46.8 | 30.7 | 11.6 | 26.0 | 53.5 | 29.3 |
| Comb | 45.5 | 7.5 | 14.1 | 34.5 | 36.1 | 49.9 | 33.8 | 11.3 | 26.3 | 55.8 | 31.5 |
| Ours | **88.4** | **76.6** | **80.6** | **90.9** | **91.6** | **95.9** | **69.7** | **81.3** | **89.1** | **93.1** | **85.7** |

## 4.5 Attack the Animals Classifier

We evaluated the attack effect of our method on ten-animal classification tasks. We chose Midjourney as the generator of adversarial images. and the settings for adversarial prompt structure and word space were introduced in Section 3.2. For the target animal, we used 10 types of animals in Animals-10. We used our adaptive GA method to get the adversarial prompts. For each target animal, we initialized 20 prompts with random word initialization. The probability of mutation was 0.01, and the hyperparameter $\lambda$ in the fitness function was 0.1. The termination condition was that the number of iterations reached 8 generations. For fair comparison, we chose three methods, clean image generation (e.g. the prompt is "generate an image of dog"), random word selection and combinatorial

testing [28] as control experiments. Under each setting, we got 20 prompts for each target animal, and each prompt generated 8 images through Midjourney, so a total of 160 images for each target animal were generated under each setting.

We inputted these images into the animal classifier ResNet101 which was trained on ImageNet, and calculated the ASRs. The results are presented in Table 1. It indicates that, on the 10-animals classification task, our method achieved an average ASR of 84.7% for the ResNet101 classifier. In contrast, the average ASR for clean image generation, random word selection and combination testing was 4.1%, 36.5%, and 37.6%, respectively. Examples of adversarial prompts and images are shown in *SM*. We observed variations of baselines and attack effects for different animal categories. For example, the ASRs of clean image generation for sheep and dog

**Table 3: Attack Stability Verification.**

| Times | 1 | 2 | 3 | 4 | 5 | 6 | 7 | 8 | 9 | 10 | Average |
|---|---|---|---|---|---|---|---|---|---|---|---|
| ASR(%) | 87.5 | 77.5 | 85.0 | 92.5 | 87.5 | 75.0 | 90.0 | 82.5 | 82.5 | 85.0 | 84.5 ± 5.4 |

were 0.0% and 29.2%, which varied a lot. The reason may be as follows. As stated in Section 4.2.2, there is a category imbalance problem in ImageNet, which may cause unbalanced classification performance of classifiers trained on ImageNet and attack effects for different categories. Despite this, the ASRs of our method for different animals were all higher than that of control experiments, which indicates the effectiveness of our method.

To build a more category-balanced classifier as the attack target classifier, we finetuned the classifier ResNet101 on a category-balanced dataset Animals-10. We then attacked the finetuned classifier ResNet101, and the results are shown in Table 2. The average ASR of our method was 85.7%, which was much better than that of clean image generation (1.0%), random word selection (29.3%) and combination testing (31.5%). This further indicates that our method is effective. Figure 3 shows a set of examples.

### 4.6 Stability of the Attack

Since the generation of text-to-image models is a stochastic process, the same prompt may lead to different images in successive queries. To verify the stability of our attack method, we selected 10 optimized prompts, then inputted it into Midjourney 10 times, and tested the ASRs of the 40 generated adversarial images each time. The results are shown in Table 3. It indicates that the ASRs of our adversarial prompts were higher than 75% in 10 successive attacks. The average ASR was 84.5% ± 5.4%. This shows that our attack method has good stability. Moreover, This suggests that, to a certain extent, our method can find the key semantic information in the natural language space, and adversarial images with such semantic information have stable adversarial effects.

### 4.7 Analyzing Adversarial Images from a Natural Language View

We tried to explore a novel perspective by analyzing adversarial images from the viewpoint of natural language. We analyzed 198 adversarial (misclassified) images and their prompts with ASR higher than 87.5% from experiments in Section 4.5 and found that the frequency of some words in these prompts were significantly higher than that of other words. For example, for "<number>", "two" appeared most frequently, and its frequency was 50.5%. For "<color>", "green" had the highest frequency, which was 61.0%. For "<weather>", "foggy" and "humid" appeared most frequently, where the frequency was 46.3% and 35.5%, respectively. For "<appearance>", "wearing clothes" and "wearing a pair of glasses" appeared most frequently, and the frequency was 38.1% and 35.5%, respectively. For "<gesture>", "stretching" had the highest frequency, which was 53.3%. This indicates that when the above adversarial semantic information appears, the generated images are prone to cause classifier errors.

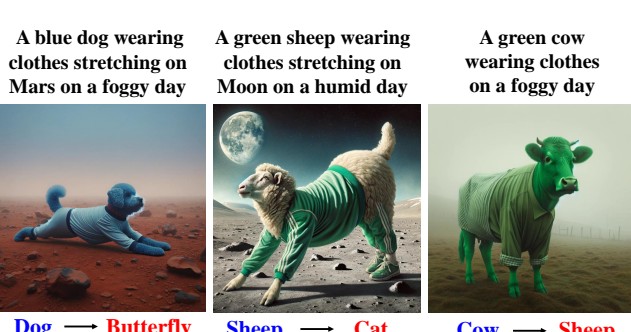

**Figure 4: Examples of generated images with adversarial semantic information for animal classification attacks.**

To verify the above conclusion, we try to combine the high-frequency adversarial semantic information such as "green", "wearing clothes", "foggy", etc. into the prompts. For example, the prompt is "an image of dog wearing clothes on a foggy day". We got 12 prompts in this way and then input them to Midjourney to generate 48 images. The generated images were input to ResNet101 classifier. The results indicate that 72.9% of the images with adversarial semantic information were misclassified, in contrast, only 29.3% of the images generated by random word selection were misclassified. Some examples of adversarial images are shown in Figure 4. It indicates that the adversarial semantic information analyzed above has an important impact on the accuracy of the classifier, which helps us to understand of the failure modes of these classifiers under natural conditions.

We found that the adversarial semantic information not only existed in generated images, but also in photos captured in real world. We searched for some photos captured in the real world on Google according to the adversarial semantic information analyzed by our method. For example, we obtained 50 images returned by Google with prompts "A cat is stretching", "A horse in a foggy day", etc. For fair comparison, we also searched 50 images by Google

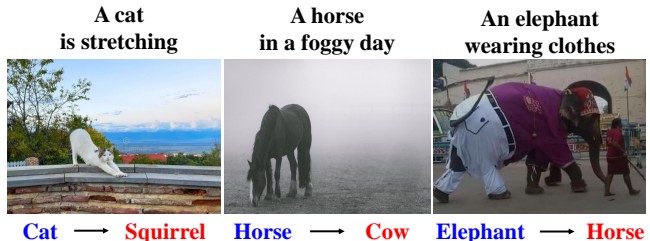

**Figure 5: Examples of Google-searched images with adversarial semantic information for animal classification attacks.**

**A black person wearing clothes on a foggy day**

**A East Asian wearing clothes in front of a brick wall on a foggy day**

**A white person wearing glasses stretching in front of a brick wall**

Black ⟶ East Asian   East Asian ⟶ White   White ⟶ East Asian

**Figure 6: Examples of generated images with adversarial semantic information for human race classification attacks.**

using prompts with random word selection as control experiments. The experimental details are described in *SM*. We input these images to the classifier ResNet101. The results show that the searched images with adversarial semantic information can also cause the misclassifications, and the ASR was 42.0%. In contrast, the ASR for random word selection was only 14.0%. Figure 5 shows some seached images with adversarial semantic information. It indicates that some semantic information in the real world (e.g. foggy, humid, stretching, etc.) may have an important impact on the accuracy of deep learning-based classifiers. This helps us to understand the weakness of classifiers implemented in real-world applications, and also helps to build more secure and robust models.

### 4.8 Zero-Shot Attack

We also found the adversarial semantic information analyzed in Section 4.7 was transferable to unseen classification tasks, and we called it zero-shot attack. We tried to apply the high-frequency adversarial semantic information obtained from animal classification attacks to attack the human race classifier. For example, the prompt is "A black person wearing clothes is stretching on a foggy day". We built 30 prompts by this way and input them to Midjourney to generate 120 images. We also set the random word selection as control experiments. The generated images were input to Vit classifier which was finetuned on FairFace dataset. The results indicated that 53.3% of the images with adversarial semantic information were misclassified, while only 25.0% of the images in control experiments were misclassified. Some examples of adversarial images are shown in Figure 6. The reason may be that some adversarial semantic information such as "stretching" and "wearing clothes" have the advantage of cross-tasks (from animal to human).

The ASR of zero-shot attacks was lower than that of our GA-based method, this is reasonable because it's a difficult task, however, it shows the possibility of transfer the adversarial semantic information to unseen classification tasks using our method.

### 4.9 Ablation Study

*4.9.1 SEM.* We conducted ablation experiments on the SEM function. We seperately used the fitness function with SEM and without SEM. The termination condition was that ASR was over 70%, and other experimental settings consistent with Section 4.5. For each

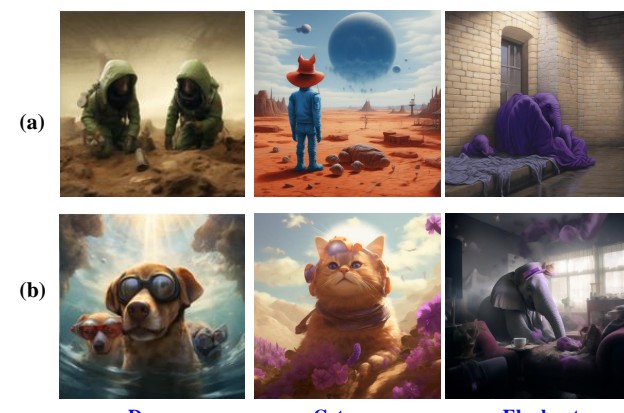

(a)

(b)

Dog            Cat            Elephant

**Figure 7: Generated images (a) without and (b) with SEM fitness function. The blue texts are the target categories.**

experimental group, we obtained 50 adversarial images. Some examples are shown in Figure 7.

We conducted a subjective evaluation and invited 10 volunteers (5 male, 5 female, ages 19-28, with normal acuity) to rate the two sets of adversarial images on a scale from 1 to 10, where higher scores indicate a greater presence of target class semantic information. The experiments were approved by the Institutional Review Board (IRB). The results showed that the average human evaluation score was 8.1±0.6 with SEM and 3.3±1.0 without SEM. It suggests that the SEM effectively enhanced the target semantic information in adversarial images while keeping a high ASR.

*4.9.2 Adaptive Word Space Reduction.* We conducted ablation experiments on Adaptive Word Space Reduction (AWSR). We seperately conducted experiments with ASWR and without ASWR. The termination condition was that ASR was over 70%, with other experimental settings consistent with Section 4.5. The results indicated that AWSR significantly reduced the number of queries (from 201 to 127) while keeping a high ASR. This not only improves search efficiency but also leads to a considerable reduction in query costs, such as the query cost for DALL·E 3 being 0.12 US dollars per image.

### 4.10 Physical Attacks

We tested the attack effect of our method in the physical world. We selected 40 adversarial images obtained in Section 4.5 which successfully misled the ResNet101 classifier. Using a Canon MF657 printer, we printed these images and then captured them with iPhone 12 Pro from a distance of 30 cm. Examples of the digital and physical images are shown in *SM*. We inputted the captured photos into the ResNet101 classifier and calculated the ASR. The results indicated that our method achieved a 100% ASR both in the digital and physical worlds. The physical world adds more perturbations [54] to the images, e.g. the printer may cause color distribution variations [15], usually leading to lower physical ASRs for previous noise-based [40] or image editing-based [56] approaches compared to their digital ASRs. However, our method are based on language with explicit semantic information, and therefore may be more robust in the physical world.

**Table 4: Attack transferability of adversarial prompts. S: source model. T: target model.**

| S \ T | Midjourney | DALL·E 3 | Stable Diffussion | DALL·E 2 | MXL | DXL | RXL |
|---|---|---|---|---|---|---|---|
| Midjourney | 93 | 63 | 73 | 78 | 80 | 90 | 80 |
| Stable Diffussion | 57 | 53 | 73 | 58 | 60 | 80 | 80 |

**Table 5: Attack transferability of adversarial images. S: source classifier. T: target classifier.**

| S \ T | ViT | VGG | ResNet | Incept | Dense | Mobile | Efficient | Squeeze | Reg | Alex | Swin | CNXL |
|---|---|---|---|---|---|---|---|---|---|---|---|---|
| ResNet | 91 | 89 | 97 | 92 | 90 | 88 | 91 | 88 | 90 | 86 | 88 | 91 |
| ViT | 95 | 93 | 84 | 99 | 89 | 89 | 96 | 93 | 93 | 89 | 92 | 88 |
| CNXL | 88 | 82 | 79 | 84 | 85 | 84 | 72 | 81 | 77 | 81 | 80 | 77 |

## 4.11 Attack Transferability of Adversarial Prompts

We tested the attack transferability of adversarial prompts of our method across different text-to-image models. Following the settings in Section 4.5, we separately optimized adversarial prompts based on a typical black-box commercial text-to-image model, Midjourney, and a typical white-box open-source text-to-image model, Stable Diffusion. For each model, we obtained 50 adversarial prompts. Subsequently, we input these prompts into various text-to-image models, including Midjourney, DALL·E 2, DALL·E 3, Stable Diffusion, Mysterious XL v4 (MXL), Dreamshaper XL alpha 2 (DXL), and Real Cartoon XL v4 (RXL), generating 200 adversarial images for each model. We then fed these adversarial images into the ResNet101 classifier, and calculated ASR.

The results are presented in Table 4. This indicates that the adversarial prompts obtained by our method can be transferred to different text-to-image models to generate adversarial images. The reason may be that some key language semantic information has an important impact on the adversarial effect. This key language semantic information can be transferred to different text-to-image models and then generate adversarial images.

## 4.12 Attack Transferability of Adversarial Images

We then evaluated the attack transferability of adversarial images of our method across different classifiers. During the optimization of adversarial images, we used the Midjourney text-to-image model and separately used a CNN-based classifier ResNet, a transformer-based classifier ViT, and an adversarial trained classifier ConvNeXt-L (CNXL) to optimize adversarial images. For each classifier, we obtained 100 adversarial images. Subsequently, we input these adversarial images into other classifiers, including ViT [12], VGG [50], ResNet [18], Inception v3 [51], DenseNet [23], MobileNet [21], EfficientNet [53], SqueezeNet [24], RegNet [45], AlexNet [27], Swin-L [38], and CNXL [7], and then calculated the ASRs.

The results are presented in Table 5, indicating the good attack transferability accross different classifiers. It is worth noting that our method successfully attacked classifiers with different architectures, including CNN-based and transformer-based architectures.

This suggests that our attack method is not entirely dependent on the classifier architecture. Furthermore, our attack method can not only attack ordinary classifiers, but also attack classifiers based on adversarial training (Swin-L and CNXL). Since traditional adversarial training usually focuses on adversarial noise, it may not be well-suited for our attack method, posing new challenges for adversarial defense methods.

## 4.13 Discussion on Potential Social Impact

As described in Section 4.8, the adversarial semantic information also exists in human race classification attacks. We also conducted the GA-based attack experiments, and the ASR against human race classifier Vit was 89%, the details are described in *SM*, which further verified the above conclusion. This revealed the potential impact of text-to-image models on social fairness. Given that many social media platforms, such as Twitter and Facebook, employ AI models for image moderation, the potential for race misclassification poses concerns for fairness. This encourage us to build more fair and robust AI models.

## 5 CONCLUSION

In this work, we propose a natural language induced adversarial image attack method, which has rich semantic information and helps humans to analyze the adversarial images from a natural language view. To adopt commercial text-to-image models for synthesizing more natural adversarial images, we propose an adaptive genetic algorithm (GA) for optimizing discrete adversarial prompts without requiring gradients and an adaptive word space reduction method for improving the query efficiency. We further used CLIP to maintain the semantic consistency of the generated images. In our experiments, we found that some high-frequency semantic information can easily cause classifier errors. These adversarial semantic information exist not only in generated images, but also in photos captured in the real world. We also found that some adversarial semantic information can be transferred to unknown classification tasks. Furthermore, our attack method can transfer to different text-to-image models and image classifiers. Our work reveals the potential impact of text-to-image models on AI safety and social fairness and inspire researchers to develop more fair and robust AI models.

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
