# OpenReview forum: "Natural Language Induced Adversarial Images"
_acmmm.org/ACMMM/2024/Conference — MM2024 Poster_

### Official Review · Reviewer_Como · 2024-05-21

**Rating:** 2
**Confidence:** 3

**Summary:**

This paper targets an interesting topic: the use of generative AI to produce adversarial examples. The authors build a framework for adversarial example generation based on the text-image generqative model, combining prompt engineering and genetic algorithm, and explore the possibilities of AI-generated adversarial examples.

**Strengths:**

+ paper is well organized
+ idea is fun

**Limitations:**

- minor grammar and typo issues, e.g., ``overal''.
- this paper lacks a solid threat model. The concept of adversarial examples (AE) in this paper needs to be redefined. Traditional AE are often based on real inputs from the physical world, and are used to deceive models through precise, human-imperceptible perturbations. The AE in this paper, on the other hand, are generated entirely by the generating AI and lack the original inputs as a reference. This significant difference leads to the fact that the traditional attack model is no longer applicable to this work.
- the lack of clarity in the definition raises many new questions: for example, how can the strength of an attack be evaluated? Does the opportunity for such an attack come from the proposed approach, or from the inefficiency of the detection model itself for generative AI? There is a need to develop a more rational attack and evaluation framework.
- The prompting framework presented in this paper is relatively simple and does not provide informative insights. Some of the adjectives observed by the authors, such as cloudy and foggy, produce more high-frequency noise, which is consistent with observations in traditional AEs.

**Suitability:**

2

---

### Official Review · Reviewer_wLGY · 2024-05-24

**Rating:** 3
**Confidence:** 3

**Summary:**

To investigate what natural conditions images are easily misled the classification method, this paper propse natural language induced adversarial attack method, where the authors leverage T2I models to generate adversarial images that mislead pretrained classifiers. Experimental results highlight the effectiveness of specific high-frequency semantic information in causing classifier errors.

**Strengths:**

- The use of natural language to generate adversarial images is a creative and novel method.
- The proposed method identifies specific high-frequency semantic information that induces classifier errors.

**Limitations:**

- I found similarities between the conclusions mentioned in Section 4.7 and some conclusions discussing the robustness of the model in image corruption. Could you please explain the difference in contribution between the attack method proposed in the paper and natural image corruption, especially, some classical image corruption methods have contained the "foggy", "rainy" and "snowy". I would raise my score if the author could make a convincing explanation.
- In my understanding, the images optimized for adversarial attacks are usually sampled from the test dataset of the victim model, and are described later with test samples (the content in brackets indicates whether it is attacked), which basically guarantees that the test sample (clean) is an I.I.D. situation, that is to say, it will be correctly classified. However, in Tables 1 and 2, I found that the samples generated by the clean method still misclassified some class. This caused me to worry about whether the clean test sample generated by the generative model might itself be an O.O.D. sample (to the victim model). Is it still meaningful to attack an O.O.D. sample? As far as I know, classification models trained on natural samples usually manifest significant performance degradation on AI- Generated samples. I hope the author can explain this. Note that this may also lead to changes in scores.

**Suitability:**

3

---

### Official Review · Reviewer_Z5Gb · 2024-05-25

**Rating:** 4
**Confidence:** 3

**Summary:**

The paper proposes a novel method for generating adversarial images using natural language prompts and simply leveraging commercial text-to-image models. They employ an adaptive genetic algorithm (GA) and an adaptive word space reduction method to optimize these prompts without requiring gradients. The results demonstrate that certain high-frequency semantic information in natural language can easily cause classifier errors. CLIP is employed to maintain semantic consistency. Experimental results demonstrate the method's effectiveness in causing misclassifications and its applicability across various models and classifiers. The authors also show semantical explainability and extensions to the proposed method.

**Strengths:**

The proposed method is simple and efficient in terms of queries. It is smart to leverage the commercial Text-to-Image models for semantical analysis and adversarial attacks.

I like the idea of using the adaptive genetic algorithm to update the words and optimize the prompt. It is a stochastic process to get a reasonable adversarial image with a natural language explanation.

The analysis and applications are very comprehensive and the authors give many use cases in extension to the proposed method.

**Limitations:**

I understand the limitation of limited queries for text-to-image generation and the authors provide an ablation study on AWSR. However, more information could be indicated to make it more convincing. How do you know there is not a combination of attributes (while individually contributes little to the attack), that could have a great impact when combined? Also, how much difference will it make in terms of ASR?
Following up with the previous comment, is the prompt of ten runs in Table 3 similar to each other?

The experiments are limited to animal and race categories, which is around 10 classes. I wonder how it would work on all the categories of ImageNet. Will it still work? If not what is the limitation?

I failed to locate how many images are used to generate ASR in a single fitness score for one prompt. Such would be important in terms of the efficiency of attacking the model. Also, the efficiency such as images generated (costs), time, or memory is missing in the paper.

There is also a series of previous works on semantic information with generative models and explainability [1-5], I wonder how the method would compare in domains other than animals in terms of semantic explanation.

[1] ICCV 2019, Semantic Adversarial Attacks: Parametric Transformations That Fool Deep Classifiers.
[2] ECCV 2020, SemanticAdv: Generating Adversarial Examples via Attribute-conditioned Image Editing.
[3] ICCV 2021, Explaining in Style: Training a GAN to explain a classifier in StyleSpace.
[4] CVPR 2023, Zero-shot Model Diagnosis.
[5] NeurIPS 2024, LANCE: Stress-testing Visual Models by Generating Language-guided Counterfactual Images.

**Suitability:**

2

---

### Meta-Review · Area_Chair_1tHc · 2024-07-01

**Recommendation:** Accept (Poster)
**Confidence:** 5

**Metareview:**

The paper proposed a method of generating adversarial images using natural language prompts. The paper received 2 borderline accept and 1 borderline reject after rebuttal. The reviewer who provided borderline reject actually raised the score from weak reject and seems to be satisfied with the rebuttal. AC looked at the reviews and rebuttal and voted for the acceptance of the paper.